# Characterization of the Role of *SPL9* in Drought Stress Tolerance in *Medicago sativa*

**DOI:** 10.3390/ijms21176003

**Published:** 2020-08-20

**Authors:** Alexandria Hanly, Jim Karagiannis, Qing Shi Mimmie Lu, Lining Tian, Abdelali Hannoufa

**Affiliations:** 1Agriculture and Agri-Food Canada, 1391 Sandford Street, London, ON N5V 4T3, Canada; ahanly2@uwo.ca (A.H.); mimmie.lu@canada.ca (Q.S.M.L.); lining.tian@canada.ca (L.T.); 2Department of Biology, University of Western Ontario, 1151 Richmond Street, London, ON N6A 3K7, Canada; jkaragia@uwo.ca

**Keywords:** alfalfa, stress, drought, miR156, *SPL9*

## Abstract

Extreme environmental conditions, such as drought, are expected to increase in frequency and severity due to climate change, leading to substantial deficiencies in crop yield and quality. *Medicago sativa* (alfalfa) is an important crop that is relied upon as a staple source of forage in ruminant feed. Despite its economic importance, alfalfa production is constrained by abiotic stress, including drought. In this report, we investigate the role of Squamosa Promoter Binding Protein-Like 9 (*SPL9*), a target of miR156, in drought tolerance. Transgenic alfalfa plants with RNAi-silenced *MsSPL9* (*SPL9*-RNAi) were compared to wild-type (WT) alfalfa for phenotypic changes and drought tolerance indicators. In *SPL9*-RNAi plants, both stem thickness and plant height were reduced in two- and six-month-old alfalfa, respectively; however, yield was unaffected. *SPL9*-RNAi plants showed less leaf senescence and had augmented relative water content under drought conditions, indicating that *SPL9*-RNAi plants had greater drought tolerance potential than WT plants. Interestingly, *SPL9*-RNAi plants accumulated more stress-alleviating anthocyanin compared to WT under both drought and well-watered control conditions, suggesting that *MsSPL9* may contribute to drought tolerance in alfalfa, at least in part, by regulating anthocyanin biosynthesis. The results suggest that targeting *MsSPL9* is a suitable means for improving alfalfa resilience towards drought conditions.

## 1. Introduction

As the world population increases, so does the demand for resources to support it. For example, global crop demands are projected to double by 2050 [1]. There exists a number of strategies to meet crop demands including increasing farmable land and improving productivity of existing farmland [1,2]. Unfortunately, some strategies that create new agriculturally available land, such as clearing, also result in the release of greenhouse gases, contributing to climate change [1]. Climate change can also cause extreme environmental conditions, such as drought, that must be overcome to achieve maximum agricultural production [3]. Areas such as the Canadian Prairies have seen a decrease in annual precipitation [4], leading to drought conditions that result in the reduction of crop yields [5]. Investigating and enhancing existing predispositions for tolerance to water deficiency could thus provide tools for crop improvement to secure the global food supply.

*Medicago sativa* (alfalfa) is a perennial legume grown on 3.8 million ha in Canada [6]. Alfalfa is an important staple in livestock nutrition due to its large supply of protein [7,8]. Its status as an important forage crop in Western Canada prompted careful evaluation of the trends in alfalfa yield in response to environmental variations [9].

In order to minimize water loss and to increase water uptake, plants respond to drought through physiological changes that include increasing root length, closing stomata, and delaying leaf senescence [10,11,12]. Plants can also utilize soluble secondary metabolites and inorganic compounds, known as osmolytes, to maintain cell turgor and hydrostatic pressure under drought [13]. Drought-sensitive plants can ultimately succumb to oxidative stress due to the build-up of reactive oxygen species (ROS) that cause changes in signalling cascades and transcription factors, leading to cell death [14,15]. Enzymatic ROS scavengers like Catalase 1 (CAT1) can mitigate ROS build-up and can increase drought tolerance of a plant [14,15,16]. Some secondary metabolites, such as flavonoids, are also capable of nonenzymatic ROS scavenging to decrease oxidative stress [14]. The ascorbate-glutathione cycle is an example of nonenzymatic ROS scavenging and of gene expression and activity of glutathione biosynthesis-related proteins such as Glutathione Synthase (GSH) increase in response to abiotic stress [17,18].

MicroRNAs have been garnering attention for their roles in regulating various aspects of development and stress response in plants [18,19,20,21,22,23,24,25,26,27]. Specifically, studies have shown that alfalfa yield as well as tolerance to heat, salt, and drought stress are improved or maintained in miR156 overexpression (miR156OE) plants [18,19,20,21,22,23]. miR156 has also been implicated in the determination of root length, branching, stem thickness, trichome density, and flowering time [19]. miR156 functions by silencing downstream genes encoding members of the Squamosa Promoter Binding Protein-Like (SPL) family [19,20,21,22,23,24,25,26,27,28]. SPLs, like miR156, regulate other downstream genes resulting in specific impacts on phenotypes and traits [21,22,26,27]. In alfalfa, at least seven *MsSPL* genes have been reported to be regulated by miR156 through transcript cleavage, including *MsSPL9* [28]. Previous investigations in alfalfa found that *MsSPL9* is downregulated in plants overexpressing miR156 [28]. While the full functional characterization of each of the seven *MsSPLs* has not been completed, a few have been studied in a number of other plant species [18,21,22,23,24,25,26,27]. A phylogenetic comparison of the Squamosal Promoter Binding Domain (SBP) of *SPL*s analysed in *Arabidopsis thaliana*, *Glycine max*, *Medicago sativa*, and *Medicago truncatula* revealed that *MsSPL9* is more closely related to *MtSPL9*, *AtSPL9*/*AtSPL15*, and *GmSPL9*/*GmSPL17* than any other *SPL* [28]. *AtSPL9* was shown to control vegetative phase transition [26] and epidermal wax synthesis [27], which are also regulated by miR156. Furthermore, enhanced expression of *AtSPL9* reduced anthocyanin accumulation and overexpression of miR156 increased expression of the *dihydroflavonol 4-reductase* (*DFR)* gene, leading to improved drought tolerance [21,29].

*AtSPL9* functions by interrupting the *DFR* transcription-activating complex, consisting of MYB proteins, bHLH factors, and WD40 proteins [29]. DFR is an enzyme involved in the biosynthesis of anthocyanins and proanthocyanidins [30,31]. Proanthocyanidins prevent pasture bloat in cattle, a common ailment in animals primarily feeding on alfalfa forage [30,32,33]. Anthocyanins contribute to pigmentation of some plant tissues and to scavenging of ROS [34,35,36]. 

This study examines the role of *MsSPL9* in drought tolerance and whether *MsSPL9* plays a similar role in alfalfa as it does in *Arabidopsis* with regard to anthocyanin biosynthesis and its involvement in the miR156-*SPL9*-DFR pathway. 

## 2. Results

### 2.1. Phenotypic Characterization of SPL9-RNAi Plants

Measurements of phenotypic traits were recorded in two- and six-month-old plants grown under favourable normal greenhouse conditions in order to evaluate phenotypes before and after flowering, respectively. Three RNAi-silenced *MsSPL9* (*SPL9*-RNAi) transgenic plants, R1, R2, and R3, were generated, and transcript levels of *MsSPL9* were evaluated in each (Figure 1). The three plants showed a range of *MsSPL9* silencing: from strong silencing in R3, low in R1, and moderate in R2. *SPL9*-RNAi plants were indistinguishable from wild-type (WT) plants in all phenotypic measurements made in two-month-old plants except for stem thickness (Table 1), where *SPL9*-RNAi had thinner stems compared to WT plants. After six months, all *SPL9*-RNAi plants showed significantly diminished plant height compared to WT (Table 2). However, this trait was the only phenotype that was consistent in all RNAi plants. R1 and R3 had decreased average internode length and increased lateral and total branching, whereas only R2 showed decreased main branching, decreased number of nodes, and delayed flowering time. The aforementioned changes to phenotype in the transgenic plants did not result in significant changes to overall yield in either shoots or roots compared to WT.

### 2.2. Response to Drought Stress in SPL9-RNAi Plants

To determine whether *MsSPL9* is regulated in response to drought in alfalfa, the *MsSPL9* transcript level was assessed in WT alfalfa plants exposed to 12 days of withholding water (Figure 2). The transcript abundance of *MsSPL9* was significantly decreased under drought compared to a well-watered control treatment. To understand the role of *MsSPL9* in drought tolerance, further drought experiments were performed on *SPL9*-RNAi and WT alfalfa plants. After 12 days of drought treatment, most plants showed signs of drought stress in the form of wilting, brown, and curling leaves (Figure 3). At this time, both WT and transgenic plants under drought had significantly decreased adaxial and abaxial stomatal conductance compared to their well-watered counterparts except in the abaxial surface of R2 (Figure 4A). *SPL9*-RNAi plants appeared to tolerate drought better than WT plants because, after 10 days of stress, wilting and leaf rolling was observed in WT plants but not in *SPL9*-RNAi plants. Evaluation of the extent of the observed tolerance to drought treatment was analysed by quantifying leaf senescence after 12 days of withholding water. While all plants exposed to drought had increased leaf senescence compared to their well-watered counterparts, drought-stressed R1, R2, and R3 had decreased leaf senescence compared to WT (Figure 4C). To investigate possible differences in their ability to grow under 12 days of drought stress, the difference between plant height before and after drought was determined. Decreased plant growth due to drought was only observed in the lowest silencers of *MsSPL9*, WT, and R1 plants, whereas the strongest silencers of *MsSPL9*, R2, and R3 plants were capable of maintaining growth under drought conditions (Figure 4B). However, maintenance in growth did not extend to overall dry weight (DW) biomass of aboveground tissues. Decreased aboveground dry biomass was observed in drought-stressed R1 and R2 plants in addition to well-watered R2 plants compared to WT in each treatment (Figure 5A). Differences in aboveground dry biomass were not observed between treatments in any plants. WT and transgenic plants were also indistinguishable between and within conditions when examining root dry biomass and length (Figure 5B and Appendix A).

### 2.3. Evaluation of Plant Water Status under Drought Stress

Plant water status can be used to shed light on mechanisms behind plant response to stress, and thus, it was used to evaluate responses of *SPL9*-RNAi plants to drought. Fresh weight (FW) of shoots and roots was compared in control and drought-treated plants. In plants with the highest *MsSPL9* transcript level, WT and R1, shoot and root FWs were decreased when comparing well-watered and drought conditions (Figure 5). Also, *SPL9*-RNAi plants had decreased shoot FW compared to WT when well-watered. WT plants, which had the highest *MsSPL9* transcript level, had decreased relative water content (RWC) between conditions whereas *SPL9*-RNAi plants could maintain RWC between conditions (Figure 6A). When comparing plants under drought stress, R1 and R2 plants had increased RWC compared to WT. However, R1 had decreased midday leaf water potential between treatments and an increase in water loss after 120 min under drought (Figure 6B,C). No changes in water loss or midday leaf water potential in response to drought were detected in the other genotypes.

### 2.4. Effect of Drought on the Antioxidant-Mediated ROS Scavenging Capabilities of SPL9-RNAi Alfalfa

The transcript abundance of the antioxidant-related genes *Catalase* 1 (*CAT*1) and *Glutathione Synthase* (*GSH*) was examined to determine whether *MsSPL9* serves to maintain the levels of these enzymes in alfalfa exposed to drought. WT alfalfa had increased abundance of *CAT1* under drought compared to control plants, but *CAT1* abundance did not change between treatments in *SPL9*-RNAi plants (Figure 7A). In fact, R2 had higher *CAT1* abundance than WT under control conditions. Similar maintenance of *GSH* transcript levels was also detected between drought and control treatments in *SPL9*-RNAi plants (Figure 7B). *GSH* abundance decreased in WT plants in response to drought, but R3 had increased *GSH* abundance under drought conditions compared to WT.

### 2.5. MsSPL9 Regulates Anthocyanin Biosynthesis

The involvement of *MsSPL9* in anthocyanin biosynthesis was first investigated by examining the anthocyanin content in *SPL9*-RNAi plants under drought and control treatments and by comparing them to WT. Under drought, anthocyanin content was visually increased in *SPL9*-RNAi plants (Figure 8A). The characteristic red hue of anthocyanins was also visible to a greater extent in well-watered *SPL9*-RNAi stems than in WT stems. Thus, anthocyanins were extracted from plant stems to determine relative quantities. Increased anthocyanin content was found in drought-stressed *SPL9*-RNAi plants (Figure 8B). R2 also had increased anthocyanin content under control conditions compared to WT. Next, we wanted to determine if *MsSPL9* impacts the *DFR* transcript level, as DFR is one of the critical enzymes in the anthocyanin biosynthesis pathway [30,31]. WT and R3 plants showed a decrease in *DFR* abundance in response to drought, whereas R1 and R2 plants were able to maintain their levels of *DFR* (Figure 8C). When considering the plants in the drought treatment only, R1 had an enhanced *DFR* transcript level compared to WT.

## 3. Discussion

The role of miR156 in plant growth and development has been well documented [19,24,25,26,27], but functional characterization of its target *SPL* genes is still lagging, especially in major crops, such as alfalfa. In this study, we found *MsSPL9* to play a role in determining plant height and stem thickness in alfalfa. Reduced *MsSPL9* transcript levels resulted in diminished stem thickness (Table 1) and plant height (Table 2) compared to WT plants, but the degree to which these traits were impacted was not proportional to the level of *MsSPL9*. For example, *SPL9*-RNAi plants displayed a gradient of *MsSPL9* silencing (Figure 1), but a similar gradient in the reduction of stem thickness and plant height was not observed. This could indicate that further regulation of these traits is not dependent on the level of *MsSPL9* under a certain threshold. *MsSPL9* may also play a partial role in developmental control over other phenotypic traits like branching, number of nodes, and flowering; however, no consistency between the *SPL9*-RNAi plants was observed. The lack of consistency between transgenic genotypes may be a result of other *MsSPL*s functioning in tandem with *MsSPL9* to exert phenotypic control. In *Arabidopsis*, *AtSPL9* is paralogous to *AtSPL15*, and Schwarz et al. (2008) [24] found that *SPL9/spl15 Arabidopsis* double mutants displayed phenotypes that were different from those of single *SPL9* and *spl15* mutants. The *Arabidopsis SPL9/spl15* double mutants had a greater number of rosette leaves and side shoots as well as later bolting and flowering than the single mutants [24]. Therefore, *MsSPL9* may be regulated in conjunction with yet to be discovered *MsSPL* paralogs in alfalfa to attain miR156-mediated developmental control.

Reducing *MsSPL9* transcript levels in alfalfa resulted in plants with improved tolerance to drought. Specifically, R2 and R3, the genotypes with the lowest *MsSPL9* transcript abundance, maintained plant growth (Figure 4B). In addition, all three *SPL9*-RNAi had decreased leaf senescence (Figure 4C) and maintained RWC (Figure 6A) in response to drought. This suggests that, despite underexpression of *MsSPL9* in R1, levels were not low enough to warrant the same tolerance against drought that R2 and R3 possessed. Support for the dependence of drought tolerance on the level of *MsSPL9* silencing could also be seen in the FW biomass of shoots and roots (Figure 5). Again, R2 and R3 were able to maintain FW between well-watered and drought-exposed treatments while WT and R1 plants had decreased FW. This is consistent with a similar finding reported for *MsSPL13*, where only *SPL13*-RNAi alfalfa plants with reduced *MsSPL13* levels but over a certain threshold showed significant drought tolerance [22].

The drought tolerant phenotypes observed in these *SPL9*-RNAi plants were only a subset of those reported in *SPL13*-RNAi plants by Arshad et al. (2017) [18], indicating that these SPLs may regulate different responses to stress. Arshad et al. (2017) [18] demonstrated that *MsSPL13* is targeted by miR156 to increase drought tolerance mainly through its involvement in regulating root architecture and water loss. *SPL13*-RNAi plants had roots that were thicker, denser, and longer than control alfalfa plants and lost less water in response to drought [18]. In this study, reduced *MsSPL9* levels did not affect root growth under control conditions or in response to drought (Figure 5B, Appendix A). Additionally, water loss was not mitigated in *SPL9*-RNAi but was actually enhanced in R1 plants (Figure 6C). Interestingly, R1, like WT, was not able to maintain growth when exposed to water-deficit conditions. Considering that R1 also had decreased leaf water potential in response to drought, the less than ideal water status of R1 plants likely contributed to its greater susceptibility to drought compared to the other *SPL9*-RNAi plants. Therefore, the differences in responses between *SPL13*-RNAi reported by Arshad et al. [18] and the *SPL9*-RNAi alfalfa plants analysed in this study suggest existence of multiple *MsSPL*s that are regulated when alfalfa is exposed to drought. 

Arshad et al. (2017) [18] also provided evidence for the involvement of miR156 in the regulation of antioxidants to combat ROS produced as a by-product of drought stress, noting an increase in total antioxidant content in miR156OE plants exposed to drought. If miR156 targets *MsSPL9* to impact the levels of antioxidants present in the plant, miR156OE and *SPL9*-RNAi plants should have similar responses. The levels of *CAT1* and *GSH* were measured to ascertain if *MsSPL9* is involved in the regulation of antioxidant activity. WT plants had enhanced *CAT1* in response to drought, but despite initially higher *CAT1* under control conditions in R2, all *SPL9*-RNAi plants maintained *CAT1* in response to drought (Figure 7A). It could be interpreted that WT plants are under greater stress and therefore upregulate *CAT1* to respond to possible enhanced ROS production, whereas *SPL9*-RNAi plants were already more tolerant to the drought conditions and thus maintained *CAT1* levels. *GSH* was also maintained in *SPL9*-RNAi plants; however, in contrast to *CAT1* levels, WT plants had reduced *GSH* in response to drought and R3 had enhanced *GSH* under drought (Figure 7B). These contradicting results require further investigation to determine if *MsSPL9* is definitively involved in antioxidant activity. 

Underexpression of *MsSPL9* increased anthocyanin content under drought stress in *SPL9*-RNAi plants (Figure 8B). This supports the conclusions made by Cui et al. (2014) [21], who found that *AtSPL9* regulates anthocyanin biosynthesis to affect drought tolerance. Thus, further investigations to confirm the direct interaction of *MsSPL9* and *DFR* was warranted. Unexpectedly, levels of *DFR* transcripts did not exactly correspond to changes in anthocyanin content (Figure 8C). *DFR* was significantly reduced in WT plants in response to drought, but no changes in anthocyanin content of WT between conditions were found. Surprisingly, *DFR* decreased in response to drought in R3 despite elevated anthocyanin content compared to WT plants under drought. In R1, enhanced *DFR* did correlate with enhanced anthocyanin content under drought. R2, however, had enhanced anthocyanin content compared to WT in both control and drought treatments but *DFR* did not change. A potential explanation for this discrepancy is that *DFR* transcription in alfalfa may be regulated through a different *Ms*SPL to impact anthocyanin biosynthesis. Feyissa et al. (2019) found enhanced levels of DFR expression in *SPL13*-RNAi alfalfa that was also tolerant to drought and demonstrated the direct interaction between *Ms*SPL13 and the *DFR* promoter region [22]. Analysis of a direct interaction between *MsSPL9* and DFR needs to be examined to test the possibility that, for example, *DFR* transcription is not regulated by *MsSPL9*. Another potential explanation for this discrepancy is that *MsSPL9* does interact with DFR in a similar fashion as was discovered in *Arabidopsis* but has a yet-to-be-discovered paralog partner, as was previously discussed. In this case, if the paralog is not silenced along with *MsSPL9*, it could be acting as the sole repressor of *DFR*, essentially making up for the absence of *MsSPL9*. Although no *MsSPL15* has yet been found in alfalfa, conflicting results regarding anthocyanin biosynthesis and *DFR* transcript abundance point towards the existence of similar acting genes. Regardless, further investigations into the role that *MsSPL9* plays in anthocyanin biosynthesis is required as it has potential to become a tool for improving alfalfa performance under drought stress.

Thinner stems and decreased plant height in *SPL9*-RNAi alfalfa demonstrated that *MsSPL9* is involved in regulating alfalfa physiology. Under drought stress conditions, *SPL9*-RNAi alfalfa also displayed increased anthocyanin biosynthesis, resulting in plants with decreased leaf senescence, increased RWC, and the ability to maintain growth. Therefore, *MsSPL9* is capable of regulating traits related to drought tolerance in alfalfa. Future studies regarding the exact mechanism through which *MsSPL9* achieves this regulation are needed, and resulting information can be applied to improve the ability of alfalfa to resist the detrimental effects of drought conditions. 

## 4. Materials and Methods

### 4.1. Plant Material

The WT *Medicago sativa L.* (alfalfa) germplasm used was propagated from clone N4.4.2 [37] obtained from Daniel Brown (Agriculture and Agri-Food Canada, London, ON, Canada). Plant material was grown under greenhouse conditions (16-h light/8-h dark, 56% humidity, 23 °C) for the duration of all experiments. WT and transgenic alfalfa plants were propagated from stem cuttings placed in Oasis Rootcubes^®^ for four weeks. Rooted cuttings were transferred to BX Mycorrhizae (PRO-MIX^®®^, Smithers-Oasis North America, Kent, OH, USA) soil in six-inch pots. Plants were watered twice per week and allowed to grow for a month before being used in any experiment. 

### 4.2. SPL9-RNAi Design and Construction

*Medicago sativa-*specific primers (Appendix A) were used to amplify a blunt-end fragment of *MsSPL9* that was cloned into the pENTR vector (Invitrogen, Carlsbad, CA, USA) according to the pENTR/D-TOPO Cloning Kit (Invitrogen). This fragment was designed to be outside the conserved SBP domain to ensure *SPL9*-RNAi specificity (Appendix A). Positive transformants were used in a LR reaction with pHELLSGATE12 according to the Gateway LR Clonase II Enzyme Mix protocol (Invitrogen). *SPL9*-RNAi transgenic alfalfa were created by transferring the construct to *Agrobacterium tumefaciens* (GV3101) by heat shock, and the resulting strain was used to transform alfalfa N4.4.2 germplasm [37]. Alfalfa transformation proceeded by soaking N4.4.2 germplasm in *A. tumefaciens*-inoculated Luria-Bertani (LB) media containing 20 μM acetosyringone and by inducing callus formation, embryo induction, and plant development as described by [38]. Positive alfalfa transformants were first selected by kanamycin antibiotic selection, potential transgenic plants were tested for the presence of the transgene by PCR analysis using primers nptII-F and nptII-R (Appendix A), and *MsSPL9* transcript abundance was determined by qRT-PCR using primers LA-Ms*SPL9*-Fq1 and LA-Ms*SPL9*-Rq1 (Appendix A).

### 4.3. SPL9-RNAi Growing Conditions, Propagation, and Characterization

Phenotypic characterization was performed on two- and six-month-old WT and *SPL9*-RNAi alfalfa with approximately 10 biological replicates of each genotype. Main branches were considered those that emerged directly from the soil, while lateral branches were considered any that emerged from main branches. The tallest main branch was used to measure plant height, number of nodes, and average internode length. Plant height was measured starting at the soil level, and average internode length was considered as the ratio between the plant height and number of nodes. Approximately halfway between the second and third nodes was used to measure stem thickness using a digital caliper (Mitutoyo, Kawasaki, Japan). Aboveground tissue was determined by decapitating the plant approximately 1 cm above the soil line, and any tissue below this point was considered roots. Tissue fresh weight (FW) was measured at the time of harvest, and dry weight (DW) was determined after the tissue was baked at 65 °C for 5 days. Root length was considered the length from the top of the root crown to the tip of the longest root. Flowering time was measured from the day cuttings were made to the first emergence of flowers.

### 4.4. Evaluation of Drought Tolerance

Drought trials consisted of one-month-old WT and *SPL9*-RNAi alfalfa, subjected to either drought or well-watered control treatments in two trials of five biological replicates. At the beginning of each trial, to normalize the amount of water each plant received, all plants were watered to pot capacity (approximately 150 mL in 6-inch pots). Drought was imposed by withholding water, while the control treated plants were watered regularly. All measurements were performed 12 days after the start of the trial. Growth during the trial was determined from the difference between plant heights before and after the trial. Leaf senescence was determined from the ratio of senesced to total number of leaves. Tissue biomass and plant height were measured using the same procedure as for *SPL9*-RNAi characterization. Analysis of roots under drought stress (root length, fresh weight, and dry weight) were performed by growing rooted cuttings in containers (Greenhouse Megastore, Danville, IL, USA) for five weeks before imposing trial conditions. The longest root was measured for length, and aboveground tissue was separated from the roots at the soil line to measure FW and DW.

### 4.5. Evaluation of Water Status

Ten leaves were used to measure relative water content (RWC) according to the following equation:(1)$$\mathrm{RWC}=\left[\frac{\mathrm{FW}-\mathrm{DW}}{\mathrm{FTW}-\mathrm{DW}}\right]\times 100$$
where FW is the weight of the tissue 12 days after water was withheld, DW is the weight of the tissue sampled at the end of each condition after five days at 65 °C, and FTW is the fresh turgid weight of the tissue after two days of incubation in deionized water [18,39,40]. Leaf water potential was measured around midday, 12 days after the trial’s initiation, using a pressure vessel SAPS II Portable Plant Water Status Console (Soilmoisture Equipment Corp., Santa Barbara, CA, USA). Midday leaf water potential was recorded when a single drop of liquid was visible from the exposed petiole [41]. Rapid water loss was evaluated by measuring the FW of decapitated aboveground tissue every half hour for three hours 12 days after the trial’s initiation. Rapid water loss was calculated using the following equation:(2)$$\%water\;loss=\frac{FW_{t_{0}}\;-\;FW_{t_{n}}}{FW_{t_{0}}}\times 100$$
where $FW_{t_{0}}$ is the initial weight of the tissue at decapitation and $FW_{t_{n}}$ is the weight of the tissue at each time point (*n*) [18].

### 4.6. Stomatal Conductance

Adaxial and abaxial surface leaf stomatal conductance were determined using a Leaf Porometer (Decagon Devices Inc., Pullman, WA, USA). Measurements were taken around midday 12 days after starting the trial. The stomatal conductance of both surfaces of two leaves in a single plant were measured. The average of the technical replicates was considered one biological replicate. Two biological replicates of each genotype were measured in four separate trials. 

### 4.7. Extraction and Analysis of Anthocyanin and Antioxidants 

Approximately 250 mg of stem tissue collected from the base of each plant was used for anthocyanin extraction as described in [42]. Absorbance of the extracts was measured at 657 nm and 530 nm using a Multiskan GO spectrophotometer (Thermo Fisher Scientific, Waltham, MA, USA). Relative anthocyanin content was calculated by subtracting the difference of absorbance at 657 nm from that at 530 nm with 60% extraction solution used as a blank. Anthocyanin content was then normalized by the weight of the starting tissue. 

### 4.8. cDNA Synthesis and qRT-PCR Analysis

RNA was extracted from 100 mg of alfalfa tissue using the RNeasy Plant Mini-prep Kit (Qiagen, Hilden, Germany) according to the manufacturer’s manual. Transcript analysis of *MsSPL9*, *DFR*, *GSH*, and *CAT1* were performed by extracting RNA from apical bud, stem, and leaf tissue, respectively. Approximately 0.5 µg of Turbo DNase (Invitrogen)-treated RNA was used to synthesize cDNA using the iScript cDNA synthesis kit (Bio-Rad, Hercules, CA, USA). *Actin-depolymerizing protein 1* (*ADF1*) and *elongation initiation factor 4A* (*elF4A*) were used as reference genes (Appendix A).

### 4.9. Statistical Analysis

Statistical analysis of the data presented was performed using GraphPad Prism software. Phenotypic trait evaluations were analysed using one-way ANOVA with Dunnett’s test for multiple comparisons between WT and *SPL9*-RNAi plants. The *MsSPL9* transcript level in WT alfalfa in response to drought was analysed using unpaired, two-tailed Student’s *t*-test. Evaluation of *SPL9*-RNAi response to drought was analysed using two-way ANOVA with Sidak and Dunnett’s test for multiple comparisons between and within conditions, respectively.

## Figures and Tables

**Figure 1 ijms-21-06003-f001:**
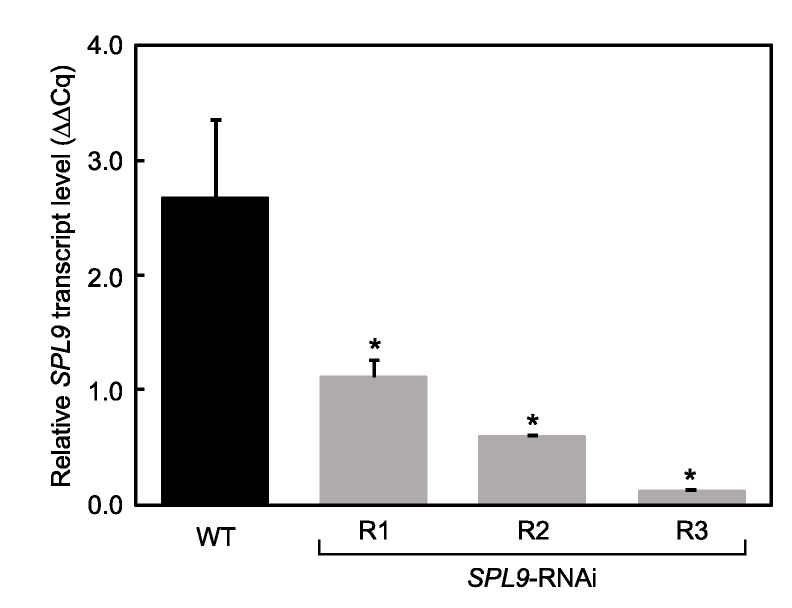
*MsSPL9* transcript abundance in RNAi-silenced Squamosa Promoter Binding Protein-Like 9 (*SPL9*-RNAi) transgenic alfalfa: the asterisks indicate significant differences between transgenic and wild-type (WT) plants (*p* < 0.05, where *n* = 3, one-way ANOVA and Dunnett’s test).

**Figure 2 ijms-21-06003-f002:**
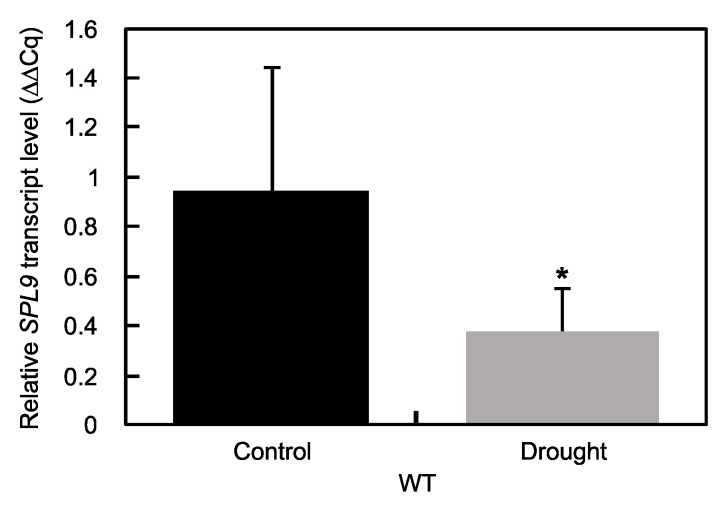
*MsSPL9* transcript abundance in well-watered control and drought-treated WT alfalfa: the asterisk indicates significant differences between conditions (*p* < 0.05, where *n* = 9, 10, Student’s *t*-test).

**Figure 3 ijms-21-06003-f003:**
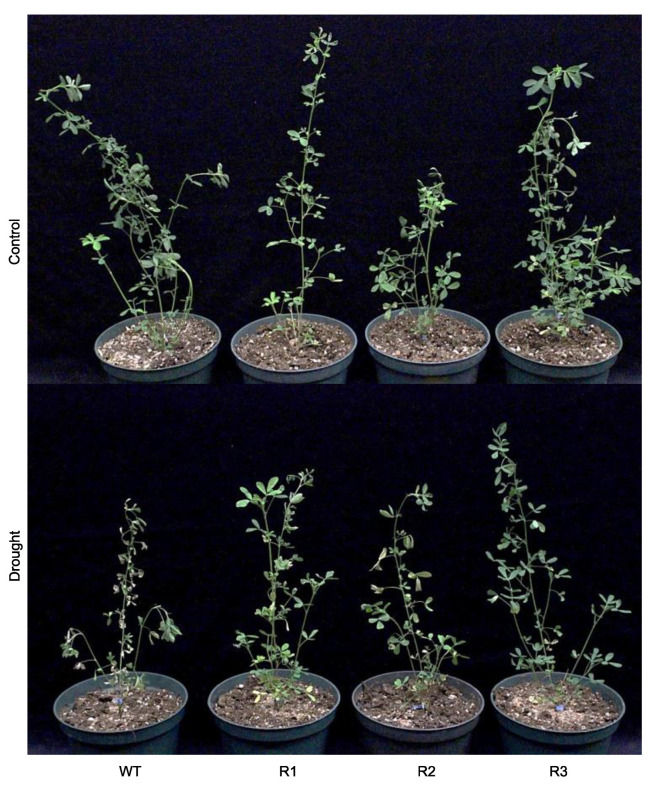
Representative *SPL9*-RNAi and WT alfalfa plants after 12 days under well-watered control and drought conditions.

**Figure 4 ijms-21-06003-f004:**
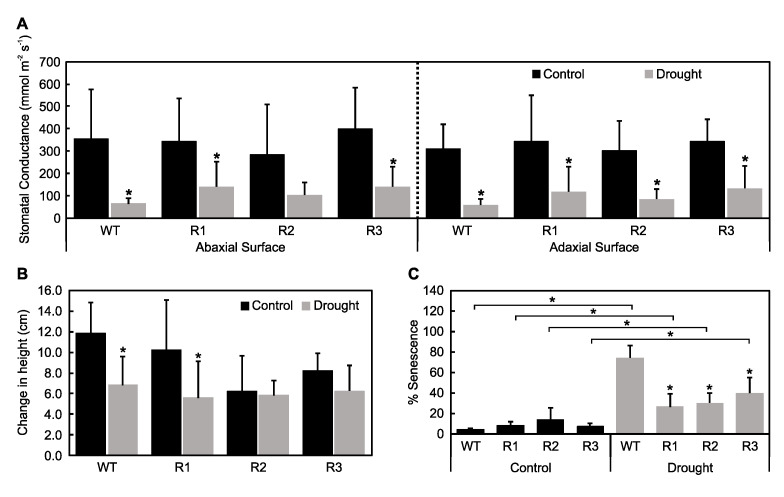
Physiological changes in *SPL9*-RNAi alfalfa under well-watered and drought conditions: (**A**) Stomatal conductance of plants measured around midday. The asterisks indicate significant differences between treatments (*p* < 0.05, where *n* = 8, two-way ANOVA and Sidak test). (**B**) Plant height difference between initiation and end of trial (12 days): the asterisks indicate significant differences between treatments (*p* < 0.05, where *n* = 10, two-way ANOVA and Sidak test). (**C**) Percent of total leaves that were senesced after plants were exposed to 12 days of control and drought treatments: the asterisks indicate significant differences within treatments (Dunnett’s test), and the bars indicate significant differences between treatments (Sidak test) (*p* < 0.05, where *n* = 10, two-way ANOVA).

**Figure 5 ijms-21-06003-f005:**
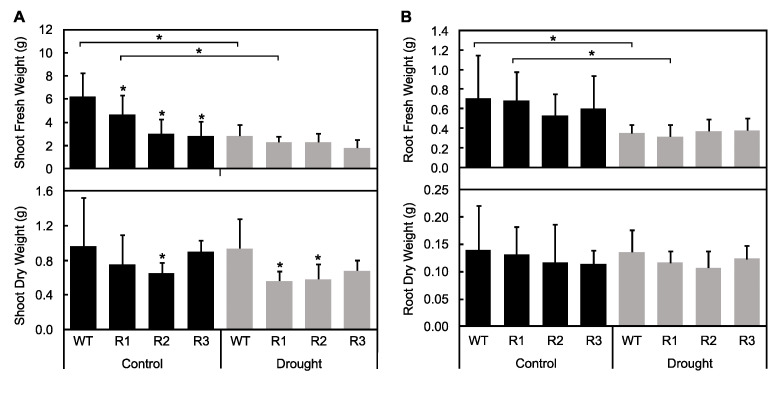
Effect of drought on biomass in WT and *SPL9*-RNAi alfalfa: (**A**) aboveground tissue fresh weight (top) and dry weight (bottom) of alfalfa plants after 12 days and (**B**) root tissue fresh weight (top) and dry weight (bottom) after 12 days of drought exposure. The asterisks indicate significant differences within treatments (Dunnett’s test), and the bars indicate significant differences between treatments (Sidak test) in a two-way ANOVA where *p* < 0.05 and *n* = 10.

**Figure 6 ijms-21-06003-f006:**
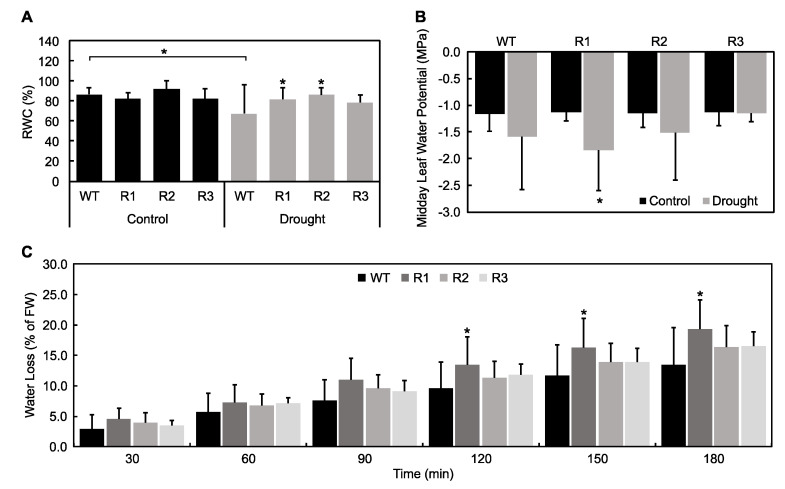
Effect of drought on plant water status in WT and *SPL9*-RNAi plants: (**A**) relative water content (RWC) of 10 leaves harvested from plants under both conditions. The asterisks indicate significant differences within treatments (Dunnett’s test), and the bars indicate significant differences between treatments (Sidak test) in a two-way ANOVA where *p* < 0.05 and *n* = 10. (**B**) Leaf water potential of plants measured around midday: the asterisks indicate significant differences between treatments (*p* < 0.05, where *n* = 9, 10, two-way ANOVA, Sidak test). (**C**) Water loss assay performed by weighing aboveground tissue every half hour for 3 h after decapitating plants that were exposed to drought: the asterisks indicate significant differences from WT within the time point (*p* < 0.05, where *n* = 10, two-way ANOVA and Dunnett’s test).

**Figure 7 ijms-21-06003-f007:**
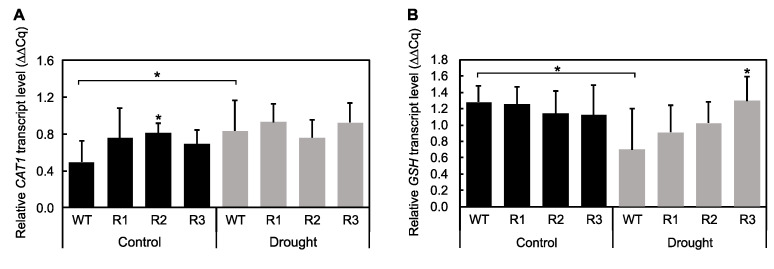
Impact of *MsSPL9* on antioxidant activity in leaves of alfalfa WT and *SPL9*-RNAi: (**A**) transcript levels of Catalase 1 (CAT1) leaves. (**B**) Transcript levels of Glutathione Synthase (GSH): the asterisks indicate significant differences within treatments (Dunnett’s test), and the bars indicate significant differences between treatments (Sidak test) in a two-way ANOVA where *p* < 0.05 and *n* = 8–10.

**Figure 8 ijms-21-06003-f008:**
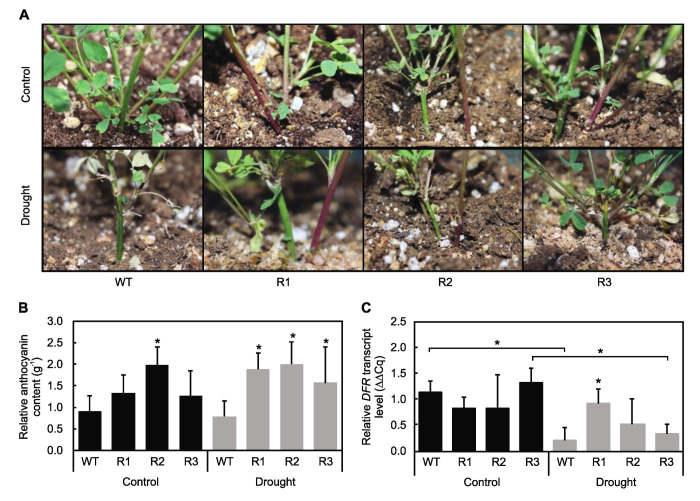
*MsSPL9* role in anthocyanin biosynthesis: (**A**) stem base phenotypes after 12 days of control and drought conditions, (**B**) relative anthocyanin content extracted from alfalfa stems (*n* = 10), and (**C**) *dihydroflavonol 4-reductase* (DFR) transcript abundance in alfalfa stems (*n* = 2–5). The asterisks indicate significant differences within treatments (Dunnett’s test), and the bars indicate significant differences between treatments (Sidak test) in a two-way ANOVA where *p* < 0.05.

**Table 1 ijms-21-06003-t001:** Physiological traits measured in two-month-old WT and *SPL9*-RNAi transgenic alfalfa: the asterisks indicate significant differences between transgenic and WT plants (*p* < 0.05, where *n* = 10, one-way ANOVA and Dunnett’s test).

Genotype	2-Month-Old Alfalfa			
	WT	R1	R2	R3
# Main branches	2 ± 0.82	2 ± 0.67	2 ± 0.47	3 ± 1.03
# Lateral branches	7 ± 4.19	9 ± 5.43	7 ± 4.86	11 ± 3.68
Total branching	9 ± 4.57	11 ± 5.87	9 ± 4.89	14 ± 3.74
# of nodes	10 ± 2.80	10 ± 3.05	10 ± 1.69	11 ± 2.92
Plant height (cm)	32.7 ± 12.79	32.4 ± 7.84	33.3 ± 5.76	36.5 ± 6.56
Average internode length (cm)	3.5 ± 1.61	3.3 ± 1.01	3.5 ± 0.64	3.5 ± 0.95
Stem thickness (cm)	1.53 ± 0.14	1.25 ± 0.16 *	1.23 ± 0.13 *	1.27 ± 0.10 *
Fresh weight (g)	3.61 ± 2.60	4.09 ± 2.98	3.23 ± 2.67	4.07 ± 2.48
Dry weight (g)	0.94 ± 0.60	0.99 ± 0.69	0.75 ± 0.56	1.11 ± 0.64
Time to flower (days)	NF	NF	NF	NF
Root length (cm)	37.9 ± 3.84	41.4 ± 9.35	38.8 ± 7.89	40.5 ± 5.01
Root fresh weight (g)	4.78 ± 3.20	6.03 ± 4.58	4.69 ± 3.69	5.94 ± 3.87
Root dry weight (g)	0.97 ± 0.31	0.96 ± 0.40	0.88 ± 0.27	1.30 ± 0.36

**Table 2 ijms-21-06003-t002:** Physiological traits measured in six-month-old WT and *SPL9*-RNAi transgenic alfalfa: the asterisks indicate significant differences between transgenic and WT plants (*p* < 0.05, where *n* = 5–13, one-way ANOVA and Dunnett’s test).

Genotype	6-Month-Old Alfalfa			
	WT	R1	R2	R3
# Main branches	27 ± 5.72	24 ± 9.52	15 ± 6.79 *	31 ± 7.78
# Lateral branches	104 ± 39.04	151 ± 31.20 *	113 ± 22.80	182 ± 34.08 *
Total branching	131 ± 40.37	175 ± 36.32 *	128 ± 20.06	213 ± 34.61 *
# of nodes	19 ± 1.48	19 ± 1.55	17 ± 0.00 *	18 ± 0.76
Plant height (cm)	86.7 ± 12.17	70.4 ± 2.75 *	71.8 ± 7.12 *	63.6 ± 3.92 *
Average internode length (cm)	4.6 ± 0.56	3.7 ± 0.22 *	4.2 ± 0.37	3.5 ± 0.18 *
Fresh weight (g)	130.62 ± 22.02	115.41 ± 22.92	101.02 ± 26.08	125.58 ± 21.71
Dry weight (g)	33.46 ± 5.61	28.55 ± 6.30	24.65 ± 6.69	33.79 ± 5.87
Time to flower (days)	114 ± 8.77	113 ± 6.18	146 ± 25.56 *	122 ± 15.55

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
