# Peer review of "Characterization of the Role of SPL9 in Drought Stress Tolerance in Medicago sativa"

_ijms, 2020, doi:10.3390/ijms21176003_

Round 1

Reviewer 1 Report

The manuscript from Hanly et al. is another piece of work that this group reports about the role of Squamosa Promoter Binding Proteins-Like (SPLs), in particular SPL9, in response to  drought, envisaging its potential use in improving alfalfa resilience to this abiotic stress. The manuscript is clearly written and the performed experiments, either related to the physiological status or to the genes and pathways involved, are satisfactory . Both studies regarding the plant characteristics and the expression of potentially SPL9-related genes  of SPL9-RNAi transgenic plants expressing different levels of SPL9 were done under well-watered and drought-stressed plants. In addition the morpho-physiological status of SPL9-RNAi plants is reported before and after flowering. The manuscript is therefore suitable for publication providing that the following minor improvements and clarification are reported.

  • Paragraph 2.2. Response to drought stress in SPL9-RNAi plants and Figure 2. I understand from the M&M that drought measures were done 12 days after imposing drought to 1 month old plants. But this is not clear in the parapgraph. Considering that also two periods (before and after flowering) are considered in the work, I suggest to include these information in the text of paragraph 2.2.
  • It is known that miR156 overexpressing plants also silence SPL9 as from Cui et al. (2014). I suggest to clearly include this specific information in the Introduction.
  • Row 113-115. “At this time, both WT and transgenic plants under drought had decreased adaxial and abaxial stomatal conductance compared to their well-watered counterparts, except in the abaxial surface of R2 (Figure 4A).” Indeed is the significance of reduction that does not occur in R2. I suggest to change with “ At this time, both WT and transgenic plants under drought had significant decreased adaxial and abaxial stomatal conductance compared to their well-watered counterparts, except in the abaxial surface of R2 (Figure 4A).”
  • Row 201: “whereas R1 and R3 plants” perhaps is not correct. It should be changed with “whereas R1 and R2 plants”

Author Response

See attached PDF file

Reviewer 2 Report

This manuscript misleads people, in which they did not present any data on miR156 and its interaction with SPL9. In this manuscript, they only study the potential effects of SPL9 in alfalfa by using RNAi to inhibit the expression of SPL9. Except this, there are several other major issues associating with this manuscript.

There are many SPL transcriptional factor genes in alfalfa, many of them are high similarity with each other. It is not clear how they only inhibit SPL9 not others. They also need to analyze the expression of all other SPL in the samples.

They should also present the expression of miR156 in all samples.

They also need to overexpress and knock out/down miRNA and study how miR156 interacts with SPL9.

Author Response

See submitted PDF file

Reviewer 3 Report

The article presents that knock-down of alfalfa SPL9, a putative target of miR159, causes a reduction of stem thickness and plant height. Also, the authors demonstrate that these SPL9-RNAi plants show a tolerance response against drought stress. They have examined various physiological changes to verify if SPL9 participates in a tolerance response in alfalfa plants. However, I think that a current version of the manuscript is not sufficient to be published in IJMS. Main and minor criticisms are as follows:

Main criticisms

  1. As described in the text by the authors, transcript levels of the SPL gene in SPL-RNAi lines are little correlated to tolerance responses against drought stress. To analyze more carefully, I recommend them to re-analyze their data set among all the samples, not just within and between treatments, statistically, because I felt that they had used weak statistical power to analyze it. The analysis may raise a new explanation about SPL9's function during drought stress. 
  2. Some responses of each SPL-RNAi line are not consistent with other responses, as described by authors. Again, a reliable statistic power will be necessary to analyze SPL9's function. 
  3. To explanation some discrepancies, they mentioned that SPL paralogs and unidentified signaling might also regulate the changes measured in this study. Even if I agree with their statement, they should have tested possible scenarios, such as expression other SPLs in SPL-RNAi plants and miRNAs,  and described their results more carefully. 

Minor criticisms

  1. I think that alfalfa SPL9 definitely regulates a stem thickness and plant height. Likely, the numbers of main branches and nodes and flowering time are strongly linked with SPL9-RNAi line2. Since drought response is not consistent with SPL9 expression, it will be better to focus on the developmental phenotypes.  
  2. 2. Line 201, .... whereas R1 and R2 (not R3). 

Author Response

See submitted PDF file

Reviewer 4 Report

The work describes the downregulation of MsSPL9 and its phenotypic effects in Medicago sativa. The work is very clearly described and conclusions are supported by the data.

I have several comments that should be addressed.

1-First there is no description of the different orthologs and paralogs of SPL9 in Medicago sativa, so it is difficult to interpret the degree of specificity of the RNAi construct.

2-A phylogenetic tree of the different MsSPL genes should be performed to see that indeed MsSPL9 is the closest gene to AtSPL9

3-The specific fragment used for silencing, its position in the coding or untranslated region is not described. Please give precise information on the construct. It could be inferred from the primers, but I think it is more transparent if it is shown

4- Please call all the MsSPL genes MsSPL throughout the manuscript, and AtSPL9 either SPL9 or AtSPL9. Otherwise it is not consistent with current nomenclature

Author Response

See submitted PDF file

Round 2

Reviewer 2 Report

Good revision and explaination